

# A structured multi-head attention prediction method based on heterogeneous financial data

Cheng Zhao[1], Fangyong Li[2], Zhe Peng[3], Xiao Zhou[4] and Yan Zhuge[5]

[1] Zhejiang University of Technology, School of Economics, Hangzhou, Zhejiang, China
[2] Zhejiang University of Technology, College of Computer Science and Technology  College of Software, Hangzhou, Zhejiang, China
[3] Zhejiang University of Technology, College of Management, Hangzhou, Zhejiang, China
[4] Zhejiang SUPCON Technology Co., Ltd, Hangzhou, Zhejiang, China
[5] Zhejiang Technical Institute of Economics, School of Digital Information Technology, Hangzhou, Zhejiang, China

## ABSTRACT

The diverse characteristics of heterogeneous data pose challenges in analyzing combined price and volume data. Therefore, appropriately handling heterogeneous financial data is crucial for accurate stock prediction. This article proposes a model that applies customized data processing methods tailored to the characteristics of different types of heterogeneous financial data, enabling finer granularity and improved feature extraction. By utilizing the structured multi-head attention mechanism, the model captures the impact of heterogeneous financial data on stock price trends by extracting data information from technical, financial, and sentiment indicators separately. Experimental results conducted on four representative individual stocks in China's A-share market demonstrate the effectiveness of the proposed method. The model achieves an average MAPE of 1.378%, which is 0.429% lower than the benchmark algorithm. Moreover, the backtesting return rate exhibits an average increase of 28.56%. These results validate that the customized preprocessing method and structured multi-head attention mechanism can enhance prediction accuracy by attending to different types of heterogeneous data individually.

## INTRODUCTION

As a vital component of the financial markets, the stock market has been the subject of substantial attention and research among scholars. The belief that individual stocks can be predicted is held by many researchers (*Babu & Reddy, 2014*), with those who are skilled in information processing being seen as having a competitive advantage in individual stock investments (*Ticknor, 2013*; *Patel et al., 2015*; *Cen et al., 2022*). The financial data of listed companies can determine the financial strength and development potential of the company and influence the long-term trend of the stock price, while technical indicators can reflect the short-term supply and demand of the stock price. The widespread adoption of the internet has resulted in most investors relying on online information for stock research, such as reading other people's comments or sharing their own opinions (*Huang*

Corresponding author
Yan Zhuge, 123198809@qq.com

*et al., 2015*). Unlike institutional investors, individual investors tend to exhibit more emotionally charged comments, greater susceptibility to market sentiment, and engage in higher frequency and higher intensity transactions, crucial factors contributing to short-term market fluctuations (*Chen et al., 2014*). Therefore, understanding market sentiment towards individual stocks during stock predictions is necessary to mitigate risk in extreme cases (*Xu et al., 2020*).

Utilizing heterogeneous financial data for stock prediction has become a new research hotspot. Technical, financial, and sentiment indicators have their characteristics. Technical data reflects the daily volatility of stock prices and is characteristic of rapid data changes. Financial data consists of the company's financial data, many of which are disclosed quarterly, with slower fluctuations. Extracting sentiment indicators from investor comment data is unstructured and difficult to extract and analyze directly. Traditional methods for processing heterogeneous data often categorize it into structured and unstructured types, without extracting more refined hierarchical features. Multiple heterogeneous financial data sources have characteristics, making it challenging to analyze when they are combined. Attention mechanism as one of the most advanced methods for temporal analysis, researchers have used attention mechanisms in stock prediction due to their ability to assign weights to different features and obtain essential features (*Niu, Zhong & Yu, 2021*). However, the presence of heterogeneous financial data increases its complexity, resulting in poor attention convergence and an inability to fully utilize the data's value.

Therefore, this article proposed a more nuanced division of unstructured and structured data, designing a data processing method customized to the characteristics of various types of heterogeneous financial data. This approach enables more effective extraction of the data's fine-grained hierarchical features. Subsequently, a structured multi-head attention mechanism generates diverse attention aggregations within different data subspaces, capturing the influence of heterogeneous financial data on stock price trends and enhancing the accuracy of stock prediction.

In summary, the contributions of this article are as follows:

(1) Based on the different types of heterogeneous financial data, customized data processing methods that are more aligned with the characteristics of the data.

(2) Propose a structured multi-head attention prediction method based on heterogeneous financial data, using the structured multi-head attention mechanism as a stock prediction model combined with technical indicators, financial indicators, and public opinion sentiment vectors for stock prediction.

(3) Compare the proposed method with traditional prediction models, demonstrate the predictive ability of the proposed model, and use backtesting to prove the model's profits in the actual market.

The rest of this article is organized as follows. "Literature Review" introduces related work and research status. "Structured Multi-Head Attention Hybrid Prediction Model" describes the proposed model in detail. "Experiment and Results" presents experimental results and discusses the results. "Conclusions" summarizes the results of the proposed model and outlines future work.

# LITERATURE REVIEW

In recent years, with the development of machine learning and deep learning techniques, some researchers have focused on predicting stocks using heterogeneous financial data. In contrast, others have introduced attention mechanisms to focus better on input vectors. This section will review the achievements and literature in these two areas in the research field.

## Analysis of heterogeneous financial data

Using multiple sources of heterogeneous financial data for the stock prediction can focus on various features. There have been many studies on using heterogeneous financial data for prediction. *Zhang, Yang & Zhou (2021)* used multiple sources of heterogeneous information such as trading data, technical indicators based on trading data, time-frequency features, and online news as data sources, combining attention mechanism with the long short-term memory (LSTM) model to extract key features from various data sources and study their impact on the stock price prediction. *Chai et al. (2020)* integrated multiple data sources such as trading data, news events, and investor comments, constructed a relationship graph to capture relevant news events of upstream and downstream products and modeled the dependence of multiple heterogeneous information sources using an extended hidden Markov model. *Zhou et al. (2023)* considered trading data, time-frequency features, technical indicators, and sentiment scores and combined Time2Vec and Transformer technologies to study the extraction and fusion of multiple heterogeneous information sources.

In heterogeneous data, financial and technical data are structured and relatively easy to process and use. However, as unstructured data, sentiment data is difficult to analyze directly. Therefore, many scholars have studied heterogeneous financial data for prediction. *Chen & Chen (2019)* integrated financial blogs and news articles. They established a dynamic public sentiment prediction model for stock prices, using extensive data analysis to effectively evaluate sentiment content and predict stock price trends. The above literature confirms the role of news sentiment in the stock market, and the influence of investors on the stock market is also increasing. *Yan et al. (2022)* proposed an Attention-based Parallel Dual-Channel Deep Learning Hybrid Model (ADDHM) that utilizes BERT to extract text semantic features, while using a CNN and BiLSTM to build a dual-channel model to extract text emotional features. *Hang & Javier (2020)* matched the corporate and social media behavior of the Chinese stock market, focusing on the relationship between stock sentiment and stock returns. Their experiment demonstrated that non-professional investors and positive sentiment have a positive effect on stock market prediction. The above research results are mainly based on micro-language analysis, lacking a macro perspective of public opinion, and the means are relatively lacking. Although *Wang et al. (2020)* and *Ma et al. (2023)* have made pioneering research results, they have not yet considered the impact of daily public opinion heat fluctuations, lack of differentiation between daytime public opinion and single public opinion, and it is difficult to reflect the complexity of market sentiment. Inspired by the above literature, this article uses public

opinion *corpus* and daily heat as unstructured data, considering the multiple aspects of the market, and then combines financial data and technical data as heterogeneous data.

## Prediction based on attention mechanism

The attention mechanism operates by computing the attention probability distribution, allowing for the identification of critical information from the inputs. This allows the neural network to concentrate on the inputs and enhance the model's performance. In recent years, due to the persistent efforts of the research community, the attention mechanism has been widely applied across various domains. A substantial body of literature on stock prediction employs the attention mechanism to enhance prediction accuracy by selectively focusing on relevant input indicators.

Many studies have observed the advantages of attention mechanisms in capturing data ability and combining various algorithms, such as convolutional neural network (CNN) and LSTM, to predict stock market performance (*Xu et al., 2022*; *Zhang, Liu & Zheng, 2021*; *Wang et al., 2022*; *Zhang et al., 2022*; *Teng, Zhang & Luo, 2022*). Attention mechanisms have demonstrated a favorable performance in handling time-series data but cannot analyze heterogeneous data. In 2017, *Vaswani et al., (2017)* introduced the Transformer model, which incorporates the concept of multi-head attention and aims to converge features from different levels. *Wang et al. (2021)* demonstrated the superiority of multi-head attention over single-head attention in performing coordinated work by applying the mechanism to locate multiple fragments of mRNA in cells. *Huang et al. (2021)* used a multi-head attention module for feature extraction and a normalized multi-head attention layer for feature convergence to tackle the keyword recognition task, showcasing the mechanism's capability in feature extraction and convergence. However, a structured grouping of the indicators has yet to be conducted. *Chen, Jiang & Sahli (2021)* Used the encoder module of the Transformer with multi-head attention to perform multi-modal affect recognition. They predicted the emotional state using multi-modal representation and enhanced the interaction between multiple modalities using multi-head attention. *Chen, Xiong & Guo (2022)* utilized multi-head attention to calculating the user's level of interest, capturing both the user's long-term and current preferences and obtaining the level of preference representation from multiple dimensions through the convergence ability of multi-head attention. In time-series prediction, the multi-head attention mechanism has also demonstrated its capability. *Yan et al. (2022)* used spatial multi-head attention to model multiple spatial patterns. Then they used temporal multi-head attention to converge the modeled spaces to enhance the modeling ability of brain functions. However, the convergence ability of multi-head attention has not yet been applied to the stock market combined with sentiment indicators. In this article, by utilizing the convergence ability of the multi-head attention mechanism, we structurally converge public opinion sentiment vectors and technical indicators of the stock into different heads, focusing on the persistent impact brought by public opinion to improve the prediction capability. As shown in Table 1, we visually compare and analyze the related work, and show the differences and connections between the proposed method and other methods.

**Table 1  Outline of related work.**

| Method | Time series analysis | Unstructured data processing | Attention mechanism | Data preprocessing |
|---|---|---|---|---|
| [17–21] | √ | × | Single head | Structured |
| [23] | × | √ | Multi-head | Unstructured |
| [24] | × | √ | Multi-head | Unstructured |
| [25] | × | √ | Multi-head | Unstructured |
| [27] | × | √ | Spatial multi-head | Unstructured |
| **Ours** | √ | √ | **Structured multi-head** | **Heterogeneously customized** |

Note:
The (✓) means that this work was done, and (✗) means that it was not.

In the table, (✓) means that this work was done, and (✗) means that it was not, and the different kinds of attention and data processing methods used by each method are shown.

# STRUCTURED MULTI-HEAD ATTENTION HYBRID PREDICTION MODEL

The framework diagram of the hybrid model proposed in this article is shown in Fig. 1.

The model comprises two modules: customized data preprocessing and a structured multi-head attention stock prediction method. The customized data preprocessing module conducts specific preprocessing on stock market technical indicators, financial indicators, and investor comments individually. It utilizes technical indicators to generate technical head data and financial indicators to generate financial head data. In the case of public opinion data, the module performs public opinion analysis by combining investor comments and popularity, resulting in the generation of corresponding public opinion vectors. The structured multi-head attention stock prediction method incorporates three attention heads: technology head, financial head, and public opinion head. It leverages the heterogeneous data generated during the data preprocessing stage and employs a structured multi-head attention prediction model based on heterogeneous financial data to predict the price trend of individual stocks.

## Customized data preprocessing

When investors express their opinions about stocks in online forums, they provide their viewpoints. The textual data used in this study is derived from the investors' comment data in the stock discussion section of the East Money website. The East Money stock discussion section is a platform where investors gather and express their comments and is one of China's largest stock communication forums. Administrators periodically delete content irrelevant to investment to ensure the purity of the forum's discussion topics. The data includes stock codes, read popularity, investor comment text, and posting time. The stock code indicates the data source in the stock discussion section, and the read popularity reflects the number of clicks and views each post has received from other investors. The investor comment text represents the title of each post, while the posting time indicates when the post was made. The structural information of the dataset is presented in Table 2.

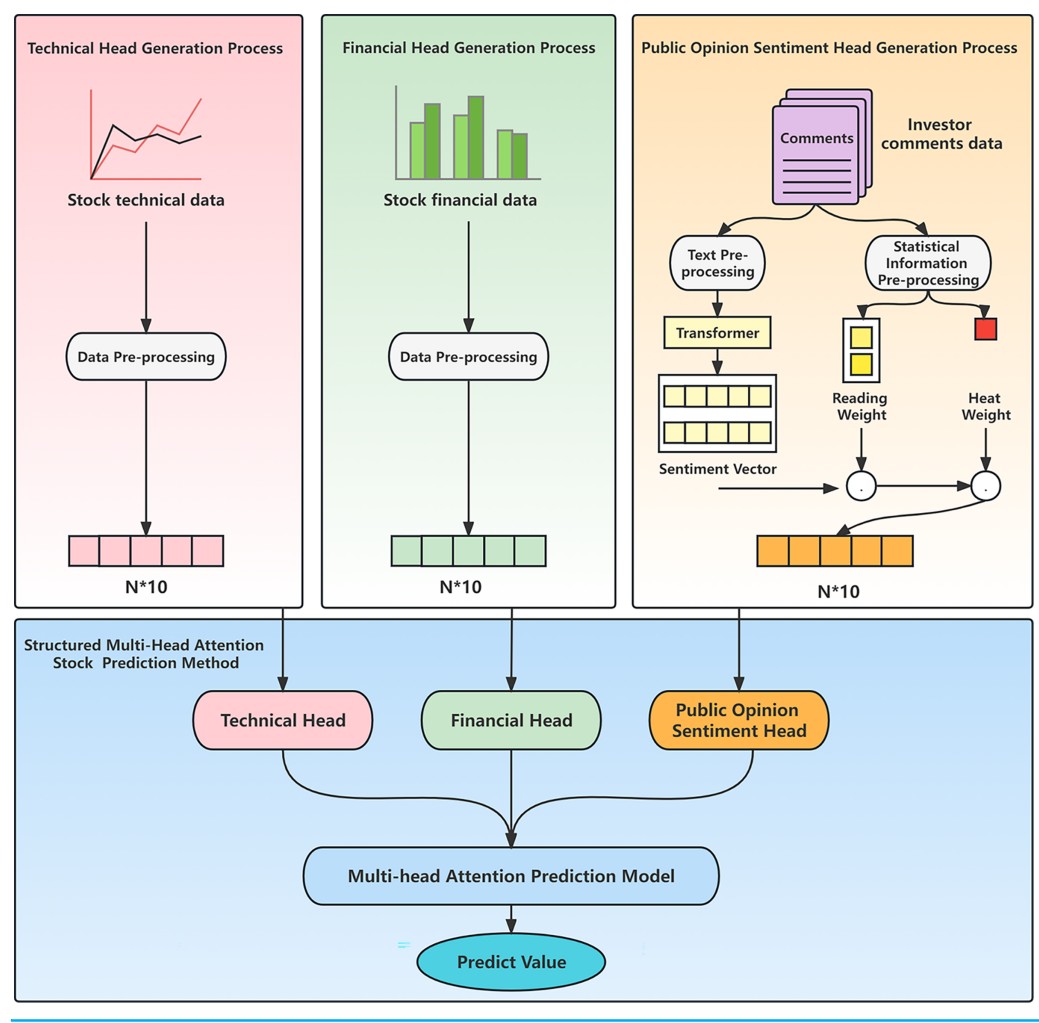

**Figure 1 Hybrid model framework.**

| Stock code | Reading quantity | Text | Release time |
|---|---|---|---|
| **Table 2** Investor commentary dataset. | | | |
| 600050 | 16,450 | What is the most terrifying? Gradual decline is the most terrifying. The market is expected to decline further. Unicom will face a difficult situation in this round. | 2019/1/3 15:59:00 |
| 600104 | 6,496 | SAIC Motor's car sales increased slightly by 1.75% in 2018. | 2019/1/4 16:44:00 |
| 600519 | 35,184 | Guizhou Province's GDP growth rate is expected to rank among the top three in the country for eight consecutive years, and local popular concept stocks continue to show positive results. | 2019/1/2 4:39:00 |

Following data purification, including eliminating invalid data such as symbols, numbers, expressions, *etc.*, the investor comments utilized in this article were segmented into individual Chinese words and punctuation marks. These segmented elements were then subjected to text tokenization, which transformed tokens into tensors suitable for training, effectively enabling the input model to recognize them as vectors.

**Table 3 Technical indicators.**

| Indicators | Meaning of indicators | Calculation formula |
|---|---|---|
| $Open_t$ | The first trading price after the opening of the SSE on the $t$-th day | – |
| $Close_t$ | The final price of the SSE on the $t$-th day | – |
| $High_t$ | The highest trading price of the SSE on the $t$-th day | – |
| $Low_t$ | The lowest price of the SSE on the $t$-th day | – |
| $Amount_t$ | The total amount traded of the SSE on the $t$-th day | – |
| $Volumn_t$ | Number of trades of the SSE on the $t$-th day | – |
| $MACD_t$ | Moving average convergence/divergence of the SSE on the $t$-th day | $2 * (DIF_t - DEA_t(N))$ |
| $RSI_t(N)$ | Relative strength indicators of the SSE on the $t$-th day | $100 - \dfrac{100}{1 + \dfrac{\text{Upward average value}}{\text{Decline average value}}}$ |
| $RSV_t(N)$ | *Raw stochastic value* of the SSE on the $t$-th day | $\dfrac{C_t - \text{Lowest price in N days} * 100\%}{\text{Highest price in N days} - \text{Lowest price in N days}}$ |
| $CCI_t(N)$ | *Consumer confidence index* of the SSE on the $t$-th day | $\dfrac{1}{0.015} * \dfrac{\dfrac{H_t + L_t + C_t}{3} - MA(N)}{\dfrac{\left| \dfrac{H_t + L_t + C_t}{3} - MA(N) \right|}{N}}$ |
| $MA_t(N)$ | Moving average of the SSE on the $t$-th day | $\dfrac{C_t + C_{t-1} + \ldots + C_{t-N+1}}{N}$ |
| $EMA_t(N)$ | Exponential average of the SSE on the $t$-th day | $\dfrac{2}{N+1} \displaystyle\sum_{k=0}^{\infty} \left(\dfrac{N-1}{N+1}\right)^k C_{t-k}$ |
| $DIF_t$ | Difference of the SSE on the $t$-th day | $EMA_t(12) - EMA_t(26)$ |
| $DEA_t(N)$ | Difference exponential average of the SSE on the $t$-th day | $DIF_t + \dfrac{N-1}{N+1} DEA_{t-1}$ |

## Technical indicators collection and pre-processing

The technical factors utilized in this study were obtained from JoinQuant. Ten technical indicators were predominantly chosen, and their specific indicators and meanings are depicted in Table 3. The first 10 indicators in the table are for experimental purposes, while the last four are explanatory. Furthermore, the data about these indicators underwent maximum-minimum normalization.

## Financial indicators collection and pre-processing

The financial factor data utilized in this study was obtained from JoinQuant. Ten financial indicators were mainly selected. The specific indicators and their meanings are depicted in Table 4. The first four indicators in the table are daily data, which is updated daily. The last six indicators are quarterly data, meaning that the data is updated quarterly with a statistical period of one quarter. Furthermore, the information on these indicators was subjected to maximum-minimum normalization.

## Sentiment indicators collection and pre-processing

The procedure for generating the Public Opinion Sentiment Vector (POS) presented in this study are described in this section. The study transforms investor comments (Text)

**Table 4  Financial indicators.**

| Indicators | Meaning of indicators |
| --- | --- |
| PE, TTM | The market price per share is a multiple of earnings per share. |
| PB | The ratio of the stock price per share to the net assets per share. |
| PS, TTM | The ratio of the stock price to the revenue per share. |
| PCF, TTM | The price-to-cash-flow (P/CF) ratio is calculated as the market price per share divided by the cash flow per share. |
| inc_return | The net asset return on equity (excluding non-recurring gains and losses) (%). |
| roa | The rate of return on total assets (ROA) expressed as a percentage. |
| net_profit_margin | Net profit margin (percentage) |
| expense_to_total_revenue | Operating cost to total revenue ratio (%)" |
| net_profit_to_total_revenue | The year-over-year growth rate of net profit percentage (%). |
| inc_net_profit_annual | The YoY (year-over-year) growth rate of net profit |

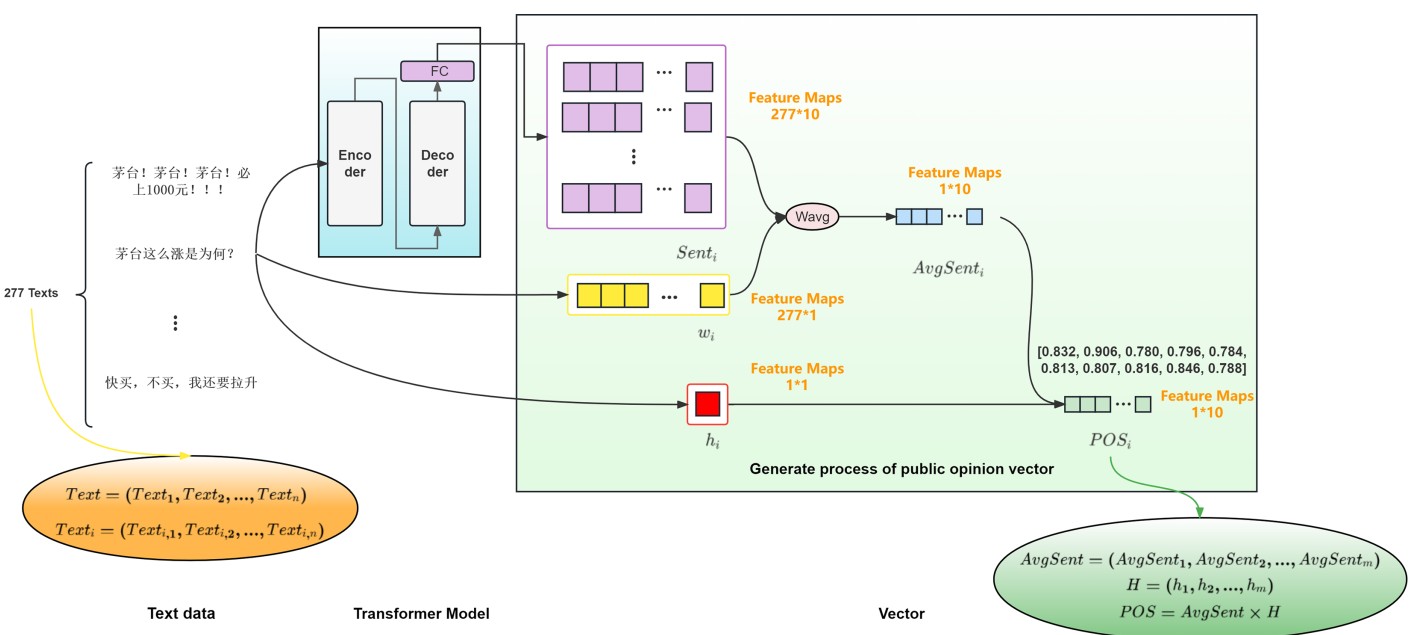

**Figure 2  Schematic diagram of public opinion sentiment vector generation for 600519 on January 15, 2019.**

into sentiment vectors (Sent) and utilizes daily reading popularity (w) and daily public opinion activity (h) as weights to generate the POS. This methodology considers the overall impact of public opinion on the market.

The proposed public opinion sentiment vector is divided according to date, and the training set has a total of m trading days. After each investor comment on the $i$-th day is converted into a sentiment vector, the public opinion sentiment vector $POS_i$ (i = 1, 2, …, m) for the $i$-th day is obtained by assigning weights to each investor comment according to its reading popularity, and the public opinion activity of that day. The process is illustrated in Fig. 2.

In Fig. 2, we use the stock with code 600519 as an example to demonstrate the process of generating the public opinion sentiment vector for the stock on January 15, 2019. Based on the opening time of the stock, *i.e.*, from 9:30 on the current trading day to 9:30 on the next trading day, the investor comments for that period are used as the investor comments that generate the public opinion sentiment vector for the i-th day. For each investor comment on the i-th day $Text_{i,j}$ (j = 1, 2, …, n), As per the technical indicator dimensions outlined in technical indicators collection and pre-processing, the sentiment vector feature dimension is determined to be 1 * 10. The sentiment vector $Sent_{i,j}$ (j = 1, 2, …, n) is computed by using a Transformer model with a multilayer Encoder-Decoder structure. Where, $Text_{i,j}$ (j = 1, 2, …, n) represents the j-th investor comment on the i-th day, $Sent_{i,j}$ (j = 1, 2, …, n) represents the sentiment vector generated by the j-th investor comment on the i-th day.

The Transformer model (*Vaswani et al., 2017*) is a novel approach to processing sequential data. It consists of multiple encoders and decoders, each designed to capture the meaning of text inputs through word vectors and position encodings. The input information comprises these two components. Each word vector represents a tensor that encapsulates the meaning of the text after pre-processing, and the position encoding captures the distance or relative position between words in a sequence. If a sentence exceeds the maximum length limit, it is truncated to that limit.

In the encoder, the vector composed of the word vector and position encoding is processed through a self-attention layer that attends to the relationships between words. This output is normalized and residual connected to a fully connected feedforward neural network, followed by additional normalization and residual connection. The resulting output is then passed to the decoder, where it is combined with the decoder input and processed through a self-attention layer.

To achieve multi-head convergence, the final input of the decoder uses a fully connected layer to extract features, and the sentiment vector dimension is dynamically determined based on the dimensions of the technical indicators in the model. This allows the sentiment vector dimension to be adapted to the technical indicator dimensions in the model. For example, on January 15, 2019, there were 277 investor comment data points in the public opinion market of stock code 600519, and a sentiment vector with a feature dimension of 277 * 10 was generated.

The reading popularity $w_{i,j}$ of each investor comment on the i-th day is obtained by normalizing the number of readings of the investor comments on the i-th day using the maximum and minimum normalization, resulting in a feature dimension of 277 * 1.

$$w_{i,j} = \frac{read_{i,j} - read_{i,min}}{read_{i,max} - read_{i,min}} \tag{1}$$

where $read_{i,j}$ represents the number of reads of the j-th investor comment on the i-th day, $read_{i,max}$ represents the maximum number of reads on the i-th day, $read_{i,min}$ represents the minimum number of reads on the i-th day, and $w_{i,j}$ represents the reading popularity of the j-th investor comment on the i-th day.

The attention of the forum users towards a certain investor comment can be understood based on the obtained reading popularity degree, which is obtained by normalizing the reading volume of the investor comments on the i-th day using the maximum and minimum normalization, resulting in a feature dimension of 277 * 1 for the reading popularity degree of each investor comment on the i-th day. The weighted sentiment vector is obtained by multiplying the sentiment vector with the reading popularity degree. The weighted sentiment vector can reflect the different impacts of investor comments with different popularity degrees. The more frequently a specific investor comment is viewed, the easier it is to influence the market sentiment and bring market fluctuations. By taking the weighted average of the sentiment vector on the i-th day and averaging it, a feature dimension of 1 * 10 for the average sentiment vector AvgSent$_i$ is obtained on the i-th day.

$$\text{AvgSent}_i = \frac{1}{n} \sum_{j=1}^{n} \text{Sent}_{i,j} * w_{i,j} \tag{2}$$

where:

$$\text{Sent}_{i,j} = \text{Transformer}\left(\text{Text}_{i,j}\right) \tag{3}$$

Simultaneously, the postings of m trading days are transformed into z-score normalized values, denoted as h$_i$, to obtain the public opinion activity. These normalized values are then utilized as weights for the public opinion sentiment vector.

$$h_i = \frac{\text{count}_i - \mu}{\sigma} \tag{4}$$

where:

$$\mu = \frac{1}{m} \sum_{i=1}^{m} \text{count}_i \tag{5}$$

$$\sigma = \sqrt{\frac{1}{m} \sum_{i=1}^{m} (\text{count}_i - \mu)^2} \tag{6}$$

where count$_i$ represents the posting volume on the $i$-th day, $\mu$ denotes the mean value of the number of investor comments, $\sigma$ denotes the standard deviation of the number of investor comments, and h$_i$ is the public opinion activity on the $i$-th day. The daily public opinion activity can be normalized through the use of a z-score standardization method, effectively eliminating the impact of excessive magnitude variations on the public opinion activity by converting it into a normally distributed form.

By multiplying the daily sentiment vector AvgSent$_i$ calculated from the investor comments of the current day with the public opinion activity h$_i$ of the current day, The daily public opinion vector POS$_i$ with a feature dimension of 1 × 10 for the i-th day was obtained.

$$\text{POS}_i = \text{AvgSent}_i * h_i \tag{7}$$

_PeerJ_ Computer Science

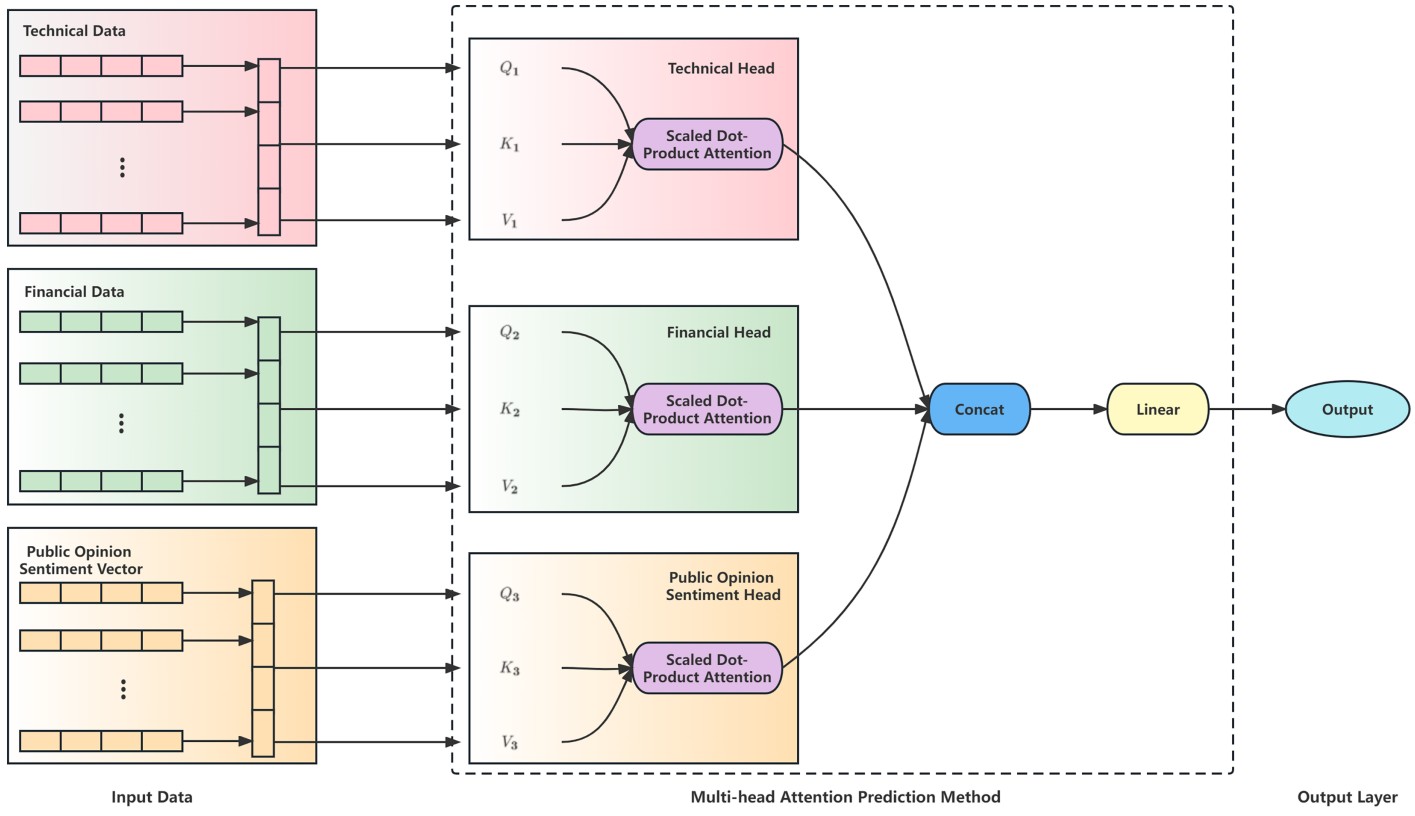

**Figure 3 Structure of structured multi-head attention stock index prediction method.**

## Structured multi-head attention stock prediction method

This section describes this study's structured multi-head attention prediction method. Traditional multi-head attention models have limited capabilities for processing heterogeneous financial data. We increase the processing capacity of such data by manually partitioning factors into technical, financial, and sentiment-related heads. In the stock prediction module, this study's structured multi-head attention prediction model is shown in Fig. 3.

We designed a structured multi-head attention prediction model based on heterogeneous financial data. First, determine the dimensions of technical and financial indicators and then generate adaptive dimension public opinion sentiment vectors based on the determined indicator dimensions. Then input the technical and financial indicators and the public opinion sentiment vectors into the multi-head attention layer to generate technical, financial, and public opinion sentiment heads. The technical head generates attention focused on technical indicators. The financial head generates attention focused on financial indicators, and the public opinion sentiment head generates attention focused on public opinion sentiment vectors. These three heads are used to aggregate data from three different aspects. After processing through the multi-head attention layer, the data is fused and input into the prediction model to predict changes in the stock.

The multi-head attention mechanism enables a group of heads to learn the different features of each position in the sequence jointly. The output of the multi-head attention module, which is composed of multi-head information, is concatenated as follows:

$$\text{Multihead}(Q, K, V) = \text{Concat}(\text{head}_1, \text{head}_2, \ldots, \text{head}_h) * W^O \tag{8}$$

where:

$$\text{head}_i = \text{Attention}\left(QW^Q_i, KW^K_i, VW^V_i\right) \tag{9}$$

$W^O$ is the weight matrix, and $W^Q_i$, $W^K_i$, $W^V_i$ are the parameter matrices of the query vector ($Q_i$), the key vector ($K_i$), and the value vector ($V_i$) in the $i$-th head. It was computed using the scaled dot product attention. The scaled dot product attention is computed as follows.

$$\text{Attention}(Q, K, V) = \text{softmax}\left(\frac{QK^T}{\sqrt{d_k}}\right)V \tag{10}$$

The $Q_i$, the $K_i$, and the $V_i$ are transformed from the input public opinion sentiment vector and technical indicators through the parameter matrix, respectively. $Q_i$ and $K_i$ are subjected to a dot product operation to obtain the relationship between the input indicators. Divide the input of the upper layer by the vector dimension $\sqrt{d_k}$, to prevent the gradient from being too small after softmax. Finally, the softmax post-weight probabilities and $V_i$ are dotted and multiplied to obtain the attention output.

Finally, the attention output is fed through a linear layer to yield the output of the linear layer, which serves as the final prediction of the stock closing price.

$$\text{Output} = \text{Linear}(W_l * \text{Multihead}(Q, K, V) + b_l) \tag{11}$$

where $W_l$ is the weight of the linear layer and $b_l$ is the bias term of the linear layer.

## EXPERIMENT AND RESULTS

In this section, we constructed the following experimental process to evaluate the proposed research model. First, examined the relationship between the number of investor comments and market volatility. Second, by taking technical indicators, financial indicators and public opinion vectors as multi-heterogeneous data inputs, compare with the benchmark algorithm. Finally, we conducted backtesting of the prediction results to demonstrate the effectiveness of the prediction method in the market.

### Experimental settings

This study selected four stocks from different industries in the Shanghai Composite Index with active market participation for the experiments. It used the data of 847 trading days from January 1, 2019, to June 30, 2022, as the source of technical and financial indicator data for the four stocks. The unstructured dataset includes 906,276 investor comments for the four stocks from January 1, 2019, to June 30, 2022.

This study's experiments divided the dataset into multiple sub-datasets using a sliding window prediction method. Each sub-dataset included a training set and test set to predict

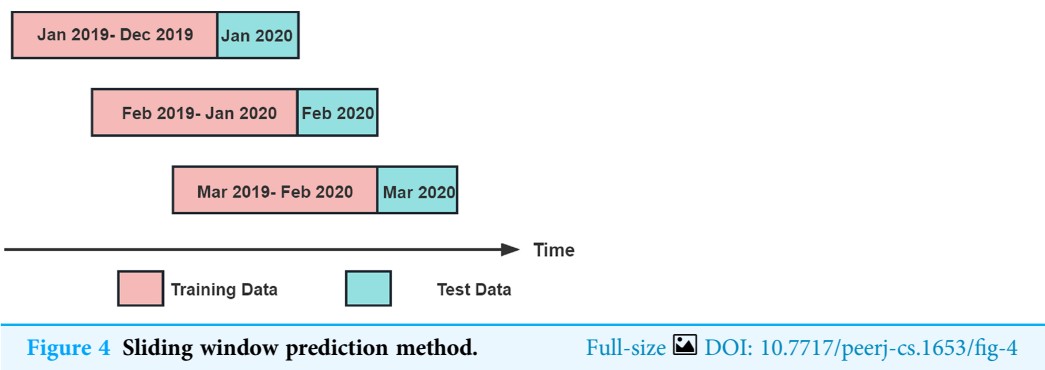

**Figure 4 Sliding window prediction method.**

the index data. The dataset is divided using a sliding window of 13 months. Each sliding window contains a training set and a test set, where the training set consists of the first 12 months of data in the window, and the test set consists of the last month of data in the window. The sliding window moves backward by 1 month each time, until it covers the entire dataset. For example, the training set of the first sliding window is the data from January 2019 to December 2019, and the test set is the data from January 2020; the training set of the second sliding window is the data from February 2019 to January 2020, and the test set is the data from February 2020, and so on, until the training set of the last sliding window is the data from June 2021 to May 2022, and the test set is the data from June 2022. The sliding window method is illustrated in Fig. 4.

## Heterogeneous data analysis experiments

This article first investigated the impact of emotional factors on prediction, and Fig. 5 illustrates the relationship between the daily number of investor comments and stock volatility.

The appearance of a large amount of text often accompanies the violent fluctuations of the stock. Although these periods account for a small proportion of the total trading time, they are when huge risks or returns are most likely to occur. At such times, investors, due to emotional fluctuations, tend to generate a large number of comment materials, which can help the model more accurately capture the market sentiment. In the case of 600519, for example, the graph shows that when investors comment more fully, the more active the public opinion is, the degree of public opinion activity is higher, such as during the period from December 9, 2021, to December 13, 2021, and from January 15, 2022, to January 21, 2022. Public opinion activity was relatively sufficient during these two periods, and market fluctuations were also more significant. Similar situations also occurred for stock 600028 from February 5, 2022, to February 15, 2022, stock 600050 from January 15, 2022, to January 25, 2022, and stock 600104 from November 4, 2021, to November 10, 2021, reflecting the high correlation between the number of investor comments and market volatility. Therefore, risk can be better managed if public opinion can be accurately analyzed.

Secondly, to verify the effectiveness of the multi-head attention mechanism in incorporating multi-heterogeneous data for stock prediction, compared the proposed model with the benchmark model. In order to compare the predictive capabilities of

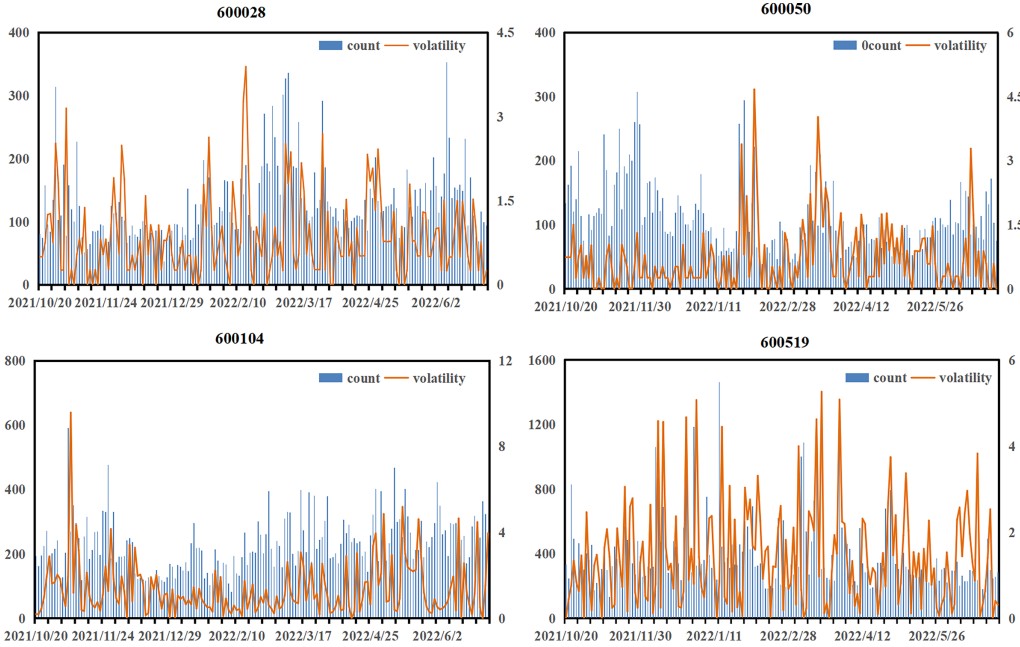

**Figure 5 Relationship between the number of daily investor comments and stock volatility.**

**Table 5 Optimal hyperparameters for each model.**

| Model | Num head | Batch size | Epoch | Learning rate | Dropout | Loss |
|---|---|---|---|---|---|---|
| LSTM | – | 128 | 200 | 0.01 | 0.2 | MSE |
| E-BiLSTM | – | 32 | 200 | 0.01 | 0.1 | MSE |
| ML-LSTM | – | 32 | 200 | 0.001 | 0.1 | MSE |
| Our model | 3 | 32 | 300 | 0.001 | 0.01 | MSE |

different models, divided the input time series into X0 (containing only technical and financial indicators) and X1 (containing technical and financial indicators as well as public opinion sentiment vectors) to verify whether different models could effectively incorporate multi-heterogeneous data and produce better prediction capabilities. To ensure a fair comparison of the predictive performance of each model, this article employed grid search to obtain the optimal hyperparameters, as shown in Table 5. In terms of the loss function, used MSE loss. For hyperparameters not mentioned in the table, used default parameters.

Figure 6 depicts the weight distribution of the structured multi-head attention mechanism in the hybrid model and displays the corresponding weight distribution plot. Each row represents the three heads of a stock, which respectively attend to technical, financial, and public opinion sentiment indicators.

It can be observed that each head generated by the model exhibits different weight distributions, reflecting distinct attention weights on technical, financial, and public opinion sentiment indicators. In the technical head, a relatively large weight is assigned to Tec[4], *i.e.*, the fifth dimension in the technical indicator, which plays a significant role in

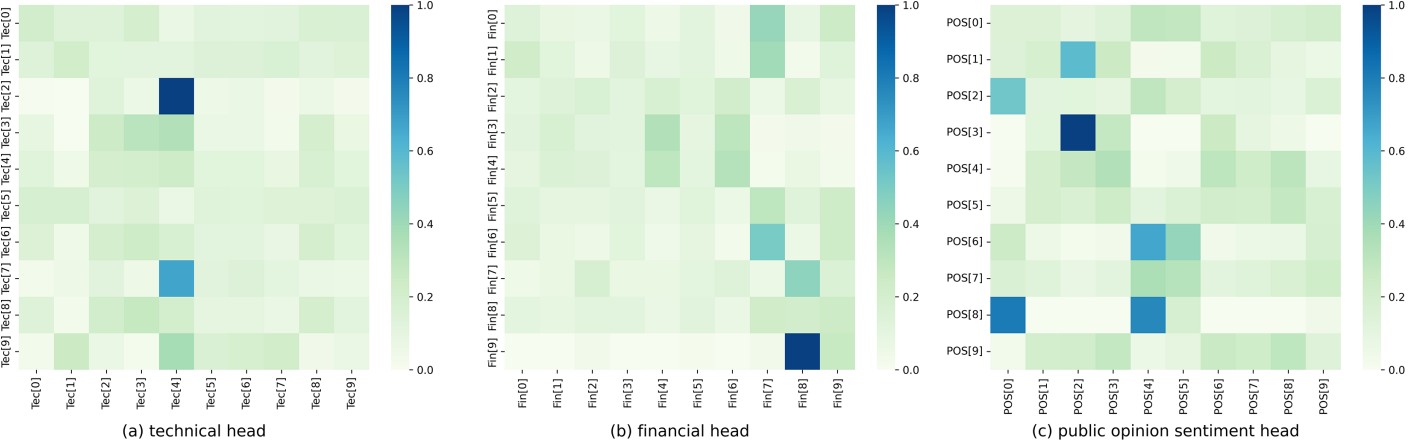

**Figure 6 Structured multi-head attention weight distribution.** (A) Technical head. (B) Financial head. (C) Public opinion sentiment head.

prediction. In contrast, the weights assigned to other dimensions are relatively weak. In the financial head, great attention is given to Fin[7] and Fin[8], *i.e.*, the eighth and ninth dimensions in the financial indicator, which have a greater impact on prediction. Additionally, Fin[4], Fin[6], and Fin[9], *i.e.*, the fifth, seventh, and tenth dimensions in the financial indicator, also receive considerable attention and influence the stock prediction. In the public opinion sentiment head, the model shows greater attention to POS[0], POS [2], and POS[4], *i.e.*, the first, third, and fifth dimensions in the public opinion sentiment vector. In contrast, the weights assigned to other dimensions are relatively even. Therefore, our structured multi-head attention model can generate different weight parameters for different heads, enabling different attention directions for the three aspects of data convergence in stock prediction.

## Accuracy analysis experiments

Figure 7 shows the box plot of the prediction errors between the predicted prices of the proposed structured multi-head attention model based on heterogeneous financial data and those of the benchmark algorithms.

The box plot visualizes each model's result distribution of daily errors, including the evaluation metrics' upper limit, lower limit, median, and outliers. The box plot shows the error distribution of various models, where the horizontal axis is the model name, the vertical axis is the error value, each diamond point represents an outlier, each bar chart represents the error range of a model, and the horizontal lines on the bar chart represent the upper limit, the upper quartile, the median, the lower quartile, and the lower limit respectively. From the box plot in Fig. 7, it can be observed that using X1 as input data, *i.e.*, combining the public opinion sentiment vector for prediction, results in better prediction performance than using X0, *i.e.*, without the public opinion sentiment vector. In the benchmark algorithms, support vector regression(SMPN) (*Xu et al., 2020*), LSTM (*Fischer & Krauss, 2018*), and evolutionary Bi-directional LSTM(E-BiLSTM) (*Zheng, Wang & Chen, 2021*), using X1 as input data generally reduces the bias and upper and lower limit

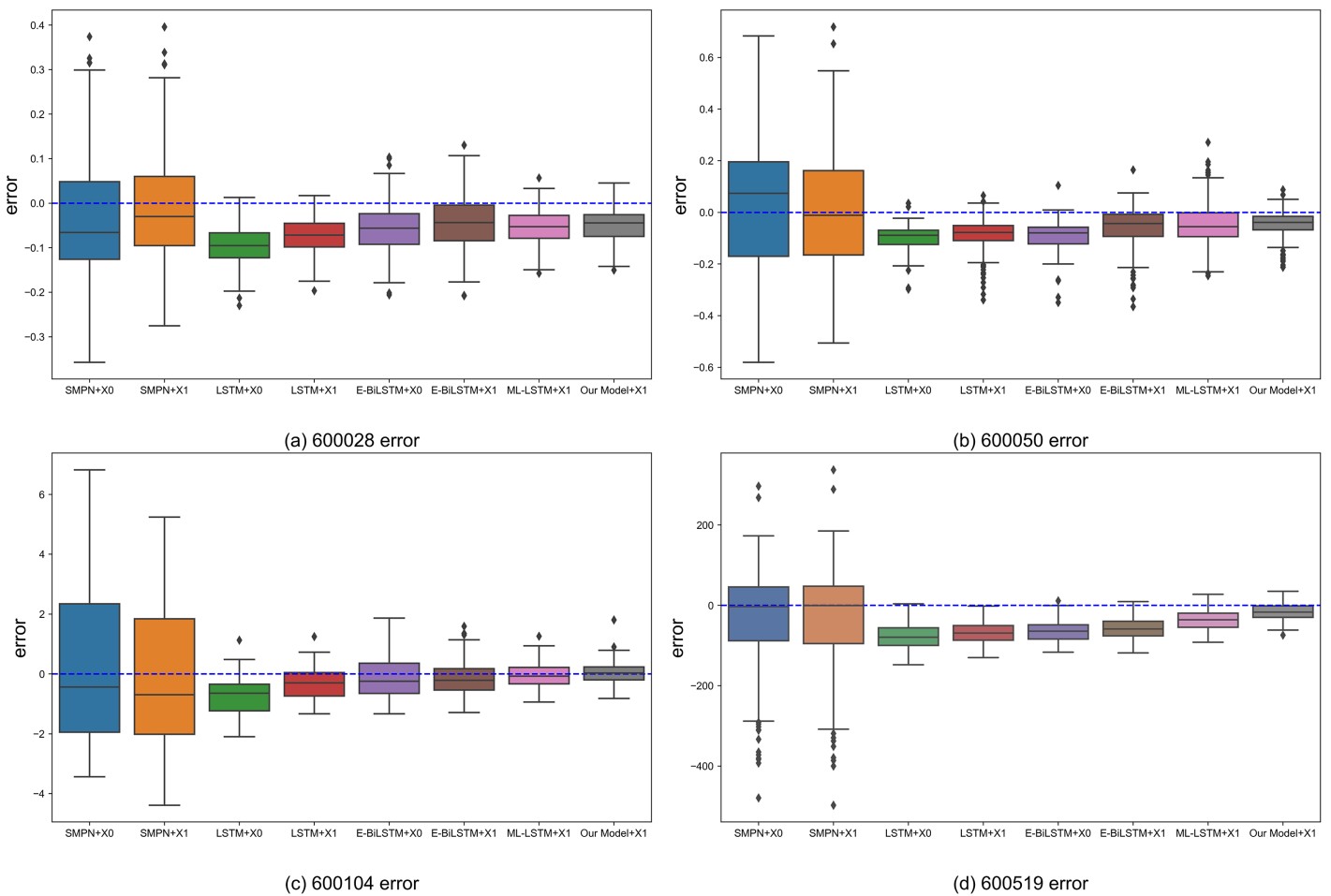

**Figure 7 Boxplot of the error value between the prediction result and the actual closing price of each algorithm.** (A) 600028 error. (B) 600050 error. (C) 600104 error. (D) 600519 error.

errors compared to using X0. The reduction in bias is more pronounced for LSTM and E-BiLSTM algorithms, whereas SMPN exhibits relatively less reduction in bias. This because SMPN cannot analyze heterogeneous data and needs to pay better attention to public opinion data. After using the attention mechanism to focus on heterogeneous data, the bias of the ML-LSTM (*Li, Shen & Zhu, 2018*) algorithm for all four stocks is reduced and closer to the upper and lower errors. The model proposed in this article obtains optimal prediction results in all four stocks, reflecting that heterogeneous data can enhance the certainty of stock market prediction and better aggregation of heterogeneous data using the model in this article to improve the prediction results.

In order to compare the accuracy of prediction results and the backtesting performance from a data-driven perspective, we selected the data of four individual stocks from October 20, 2021, to June 30, 2022, to compare the prediction and backtesting results. Constructed a timing trading strategy, generating buy signals when the model predicted a rise exceeding a threshold, and sell signals when the model predicted a drop exceeding a threshold. This

article selected MAPE, RMSE, and F-score as prediction accuracy indicators; and return rate, maximum drawdown, and Sharpe ratio as backtesting indicators. MAPE is the mean absolute percentage error used to measure the relative percentage error. The smaller the value, the better the prediction performance. RMSE is the root-mean-square error used to measure the deviation between the predicted and true values. The smaller the value, the better the prediction performance. F-score represents the harmonic mean of precision and recall in predicting classification and evaluates the effect of prediction classification. The larger the value, the better the prediction performance. The return rate represents the percentage of net profit to total capital. The larger the value, the better. The maximum drawdown represents the maximum value of the rate of return retreat when the net value of the product falls to the lowest point after a given period of time. The smaller the value, the better. The Sharpe ratio represents the excess return generated by each unit of total risk. The larger the value, the better. The comparison results between our proposed and benchmark models are shown in Table 6. Here, Buy&Hold represents the benchmark strategy of buying and holding.

Table 6 shows that utilizing heterogeneous data for predicting the four stocks reduces the MAPE on average by 0.545% and increases the F-score by 0.04 compared to the use of traditional structured data alone. This demonstrates that heterogeneous data provides comprehensive information, thereby enhancing the accuracy of prediction. Additionally, the average MAPE of the proposed method is 1.378%, which is 0.429% lower than that of the benchmark algorithm on average. This also proves the effectiveness of the customized preprocessing of heterogeneous data and then classification and aggregation through a structured multi-head attention mechanism in capturing the impact of heterogeneous financial data on stock price trends.

When comparing the backtesting indicators, the average rate of return of the proposed method is 98.731%, significantly higher than the benchmark strategy's 15.747%; the average maximum drawdown is 1.878%, significantly lower than the benchmark strategy's 24.249%; and the average Sharpe ratio is 5.544, whereas the benchmark strategy has a Sharpe ratio of only 0.76. This shows that the proposed model effectively avoids risks in the actual stock market quantification, especially during periods of active or volatile market sentiment. It can more accurately capture market sentiment, improve prediction accuracy, and enhance risk response capabilities. The strategy income formed at the end of the trading day and the daily heat map are shown in Fig. 8.

As seen in Fig. 8, the market activity levels of various stocks are different, and the active periods in the market are also different. Among them, stock 600519 has the most investment comments from market investors, and the sentiment is also the most active. The difference between the returns of our proposed method and the benchmark algorithm is most significant, demonstrating that the proposed model can produce better performance for public opinion-active markets.

In the case of 600519, for example, a number of new products were released and well received by the market during the trading day period from December 2021 to February 2022, stimulating active investor market sentiment. In the intense market sentiment, an algorithm using public opinion sentiment vectors can achieve higher returns than an

**Table 6 Evaluation of prediction and backtesting metrics for each model.**

| Stock code | Comparative experiment | MAPE(%) | RMSE | F-score | Return rate (%) | Maximum drawdown (%) | Sharpe ratio |
|---|---|---|---|---|---|---|---|
| 600028 | Buy&Hold | | | | −1.511 | −12.899 | −0.289 |
| | SMPN+X0 | 3.099 | 0.14 | 0.490 | 13.297 | −7.786 | 0.761 |
| | SMPN+X1 | 2.697 | 0.13 | 0.526 | 21.784 | −5.353 | 1.405 |
| | LSTM+X0 | 2.499 | 0.11 | 0.563 | 8.399 | −8.816 | 0.407 |
| | LSTM+X1 | 1.892 | 0.08 | 0.634 | 21.130 | −7.941 | 1.386 |
| | E-BiLSTM+X0 | 1.669 | 0.08 | 0.630 | 16.39 | −7.412 | 0.990 |
| | E-BiLSTM+X1 | 1.494 | .0.07 | 0.658 | 28.382 | −5.605 | 1.893 |
| | ML-LSTM+X1 | 1.461 | 0.07 | 0.659 | 33.567 | −6.733 | 2.432 |
| | Our Model+X1 | 1.313 | 0.06 | 0.709 | 46.682 | −3.269 | 3.663 |
| 600050 | Buy&Hold | | | | −17.541 | −19.633 | −1.507 |
| | SMPN+X0 | 5.532 | 0.24 | 0.481 | 4.929 | −7.678 | 0.169 |
| | SMPN+X1 | 4.828 | 0.22 | 0.507 | 9.237 | −7.538 | 0.493 |
| | LSTM+X0 | 2.762 | 0.11 | 0.504 | 33.132 | −3.889 | 2.296 |
| | LSTM+X1 | 2.452 | 0.11 | 0.560 | 36.991 | −6.325 | 2.723 |
| | E-BiLSTM+X0 | 2.535 | 0.11 | 0.508 | 31.418 | −3.616 | 2.330 |
| | E-BiLSTM+X1 | 1.911 | 0.10 | 0.538 | 63.215 | −6.216 | 5.166 |
| | ML-LSTM+X1 | 2.193 | 0.10 | 0.574 | 66.985 | −1.474 | 5.445 |
| | Our Model+X1 | 1.517 | 0.07 | 0.583 | 80.827 | −0.593 | 7.009 |
| 600104 | Buy&Hold | | | | −15.747 | −39.778 | −0.760 |
| | SMPN+X0 | 12.461 | 2.64 | 0.416 | 45.261 | −9.870 | 1.811 |
| | SMPN+X1 | 11.519 | 2.35 | 0.513 | 40.342 | −9.772 | 1.594 |
| | LSTM+X0 | 4.484 | 0.99 | 0.543 | 87.289 | −8.563 | 3.717 |
| | LSTM+X1 | 2.656 | 0.60 | 0.573 | 100.336 | −6.977 | 4.445 |
| | E-BiLSTM+X0 | 3.264 | 0.69 | 0.556 | 84.475 | −8.189 | 3.631 |
| | E-BiLSTM+X1 | 2.563 | 0.56 | 0.570 | 113.710 | −6.837 | 5.092 |
| | ML-LSTM+X1 | 1.770 | 0.39 | 0.571 | 114.575 | −1.978 | 5.179 |
| | Our Model+X1 | 1.505 | 0.35 | 0.583 | 127.978 | −1.920 | 6.033 |
| 600519 | Buy&Hold | | | | 9.792 | −24.686 | 0.294 |
| | SMPN+X0 | 5.049 | 135.77 | 0.428 | 27.328 | −12.447 | 1.056 |
| | SMPN+X1 | 5.004 | 135.05 | 0.454 | 34.250 | −12.500 | 1.395 |
| | LSTM+X0 | 4.326 | 84.60 | 0.522 | −0.583 | −18.271 | −0.136 |
| | LSTM+X1 | 3.758 | 73.35 | 0.599 | 30.072 | −14.584 | 1.229 |
| | E-BiLSTM+X0 | 3.545 | 69.58 | 0.615 | 40.690 | −13.631 | 1.627 |
| | E-BiLSTM+X1 | 3.208 | 63.67 | 0.608 | 63.515 | −8.389 | 2.786 |
| | ML-LSTM+X1 | 2.085 | 44.04 | 0.638 | 62.545 | −12.016 | 2.845 |
| | Our Model+X1 | 1.175 | 25.95 | 0.658 | 136.407 | −1.720 | 7.470 |

algorithm that does not use public opinion sentiment vectors, demonstrating the effectiveness of the attention mechanism that recognizes the intensity of market sentiment. From April to May 2022, market sentiment weakened. However, our proposed structured multi-head attention model based on heterogeneous financial data paid attention to

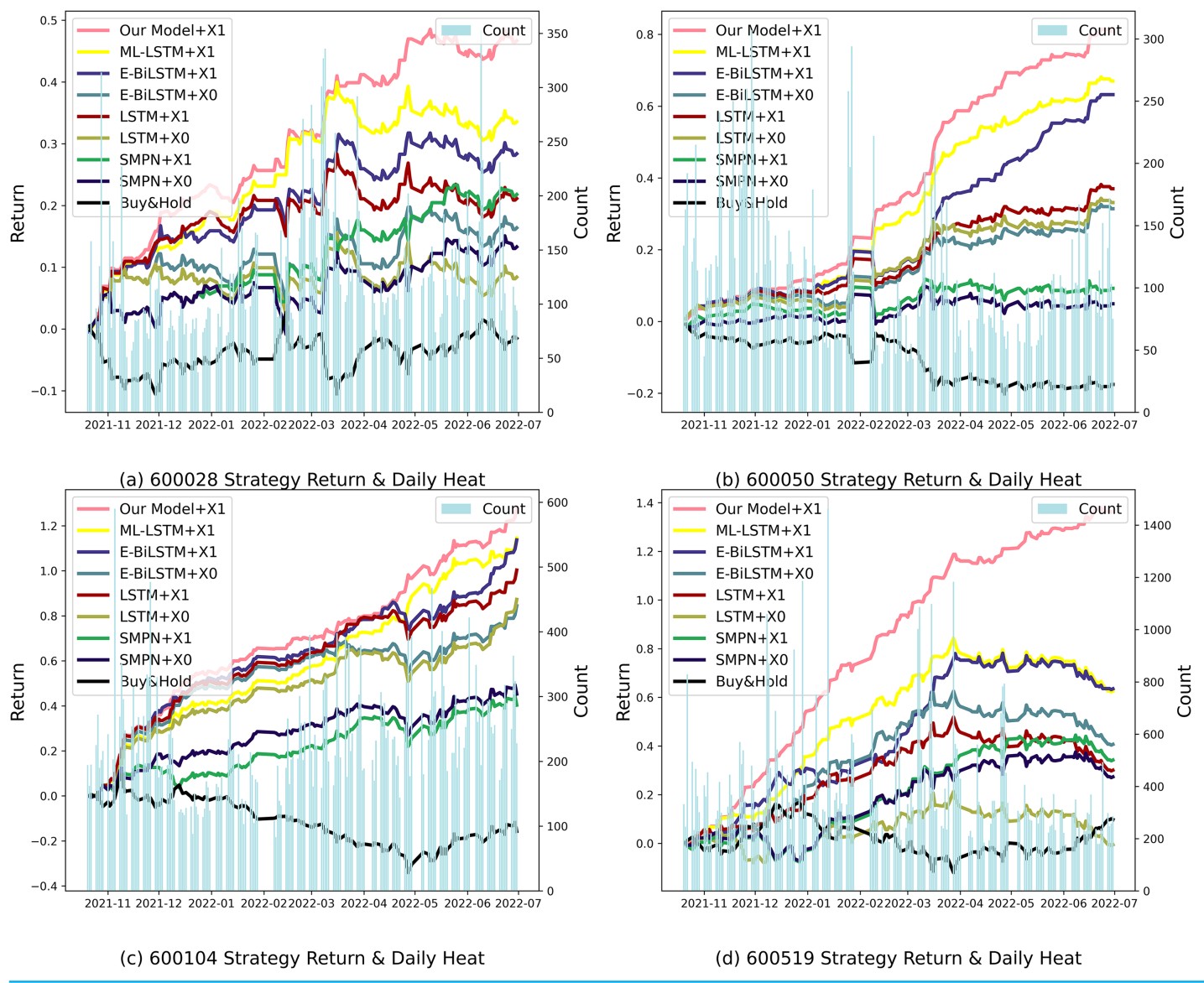

**Figure 8  Backtesting returns of different individual stock strategies.** (A) 600028 strategy return and daily heat. (B) 600050 strategy return and daily heat. (C) 600104 strategy return and daily heat. (D) 600519 strategy return and daily heat.

market sentiment fluctuations, reducing losses and demonstrating better risk management capabilities. Similar situations occurred with the stocks 600028 from May to June 2022, 600050 from April to May 2022, and 600104 from May to June 2022, and our algorithm demonstrated corresponding advantages in each case.

Our proposed model can better combine heterogeneous financial data from technology, finance, and public opinion sentiment to achieve the best prediction results and backtesting strategy returns in stock prediction.

## CONCLUSIONS

In this article, we propose a structured multi-head attention model that utilizes heterogeneous financial data for stock prediction. The model divides unstructured and structured data into finer granularity and employs customized data processing methods customized to the characteristics of different types of heterogeneous financial data to enhance feature extraction. Subsequently, the model utilizes various heads within the structured multi-head attention mechanism to generate diverse attention aggregations in three data subspaces. This allows for simultaneous extraction of technical, financial, and sentiment indicators, capturing the influence of heterogeneous financial data on stock price trends and enhancing prediction accuracy. The experiment compares the proposed model with four benchmark algorithms using related experiments conducted on four individual stocks. The evaluation metrics include MAPE, RMSE, F-score for prediction accuracy, and return rate, maximum drawdown, and Sharpe ratio for backtesting. The evaluation demonstrates the superior performance of the proposed method. Additionally, the experiment yields insightful results: (1) Strong correlation between unstructured sentiment data and market volatility is established. (2) Customized processing methods for different data lead to more accurate predictions. (3) Structured multi-head attention processing of heterogeneous financial data produces diverse attention aggregation results, reflecting various characteristics of the target stock. Consequently, the proposed method effectively handles the heterogeneity of actual stock market data and provides accurate prediction results. It also offers valuable trading suggestions in practical investment applications, aiding investors in making informed decisions.

Nevertheless, this research has certain limitations. Various country-specific factors exist in different countries' stock markets (*Kang, 2004*). Developed countries' financial markets, having more institutional customers, are less influenced by public opinion, whereas emerging markets are more susceptible to public opinion. The hybrid prediction model presented in this article is specifically tailored for Chinese *corpus* analysis and requires further adaptation for application in other countries' stock markets. This is an area we plan to explore extensively in future research.

### Funding

This research was funded by the Major Humanities and Social Sciences Research Projects in Zhejiang Higher Education Institutions, Grant Number 2023QN082 and the National Natural Science Foundation of China, Grant Number 61902349. The funders had no role in study design, data collection and analysis, decision to publish, or preparation of the manuscript.

### Grant Disclosures

The following grant information was disclosed by the authors:
Zhejiang higher education institutions: 2023QN082.
National Natural Science Foundation of China: 61902349.

## Competing Interests

Xiao Zhou is an employee of Zhejiang SUPCON Technology Co., Ltd.

## Author Contributions

- Cheng Zhao conceived and designed the experiments, analyzed the data, performed the computation work, prepared figures and/or tables, authored or reviewed drafts of the article, and approved the final draft.
- Fangyong Li conceived and designed the experiments, performed the experiments, performed the computation work, prepared figures and/or tables, authored or reviewed drafts of the article, and approved the final draft.
- Zhe Peng conceived and designed the experiments, performed the experiments, analyzed the data, performed the computation work, prepared figures and/or tables, authored or reviewed drafts of the article, and approved the final draft.
- Xiao Zhou analyzed the data, prepared figures and/or tables, and approved the final draft.
- Yan Zhuge conceived and designed the experiments, authored or reviewed drafts of the article, and approved the final draft.

## Data Availability

The relevant experimental data, including contains investor comments sourced from the East Money stock bar (http://guba.eastmoney.com/), and stock market trading data sourced from the JoinQuant quantitative trading platform (https://www.joinquant.com/data) is available in the Supplemental Files. There is Chinese content in the data, including the comments, which has not been translated into English in order to ensure the completeness of the experiment.

The experimental code, including the stock market prediction code, and the code for generating sentiment vectors from public opinion is available in the Supplemental Files.

## Supplemental Information

Supplemental information for this article can be found online at http://dx.doi.org/10.7717/peerj-cs.1653#supplemental-information.

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
