# Peer review of "A structured multi-head attention prediction method based on heterogeneous financial data"

_PeerJ Computer Science, doi:10.7717/peerj-cs.1653_

## Round 0.1 · original submission · Major Revisions

Thanks for your submission. Please incorporate the reviewers' comments. Also, make the following changes:
a) Improve introduction by making problems explicit and also making contributions clear.
b) Literature review: This section needs visuals or tabular analysis of related work.
c) Take advantage of highlighting the industrial applications of your work (maybe you can do this in the discussion or implication section). Also mention how your paper addresses the gaps that you have identified before.
d) Concluding remarks must be improved.

Reviewer 1 ·

Basic reporting

Use of investor’s sentiment, attention mechanism in financial forecasting is very common now a days. What is the novelty of the study then?
The description about the proposed hybrid model is not sufficient. How the hybrid forecast formed must be discussed clearly.
A comparative analysis with state-of-the-art forecasting models is suggested. Neural network-based forecasting models may be considered for comparison since they have good approximation ability.
The figures need to be improved for good quality.

Experimental design

Experimental study is not sufficient to establish the superiority of the proposed approach. Please conduct an exhaustive experimental work and report the analysis results.

Validity of the findings

The abstract and conclusions may be rewritten in a better way.

·

Basic reporting

Good narration and fulfilled the basic reporting when adding references for public opinion sentiment related papers can be provide wholistic of the literature. (https://www.sciencedirect.com/science/article/abs/pii/S0952197622004389)

LineNo: 458 - 600050 from April to May 2022, and 600104 from May to June,
Is these Raw data shared for evaluation??
Table1 is given in chinese will it necessary ?? if not then provide in english

Experimental design

Good to see that data collection of stock and its related pre processing in the Raw data attached in zip whereas sentiment related data and its interpretation might be missed.
In Fig.1 public opinion sentiment data is an synthesized data??
If so what might be the metrics considered it?
Can narrate it in the paper.

All the graphs needs to be correlated with outcome of the paper while making statements in line:334-336 like way other fig.s also ensure.

Validity of the findings

Validity of the findings are seems to be novel and it will be statistically sound but if different country different data how it may behave and what might be the directions ??
Which can be given in the paper.

Additional comments

If they fulfill above comments , the paper can be shaped into journal requirements.

Reviewer 3 ·

Basic reporting

Basic Reporting:
The paper is generally well written and organized. There are clear sections for introduction, methodology, experimentation and results, conclusion, and acknowledgments. The language used is clear and the technical terms are defined adequately. There are also appropriate references to support the research context. However, there are areas where further clarification is needed.
For example, the paper could benefit from a clear explanation of the sentiment vector used, how it was formed, and why it was chosen.
Figures and tables are used effectively to present results and comparison data. However, there could be more detailed descriptions accompanying each of these, such as explaining the metrics used in the box plots, or a more detailed analysis of Table 5.

Experimental design

The experimental design appears to be sound, with a clear data collection period and a well-defined benchmark for comparison.
The study's premise is well justified, and the approach to prediction using sentiment analysis is innovative. However, there could be more clarity about how the four stocks were selected and if these choices affect the study's generalizability.

Moreover, the paper would benefit from further explanation about how the multi-head attention mechanism was integrated into the prediction model, how it works, and why it was selected over other potential methods.

Validity of the findings

The findings seem valid and supported by the provided data. There are clear indicators of improvement in prediction accuracy when using the proposed model, compared to the benchmarks. However, there are a few concerns:

a. The paper suggests that incorporating sentiment analysis can result in better prediction performance. However, the authors need to ensure that the improvement is statistically significant, not merely a product of random variation.

b. The generalizability of the findings to other stocks or markets could be discussed more. The paper focuses on four individual stocks, and it would be interesting to see if the same approach would work on a broader scale or in different markets.

Additional comments

Overall, the paper presents an interesting approach to stock price prediction and provides convincing evidence that the proposed model works well for the selected stocks. However, to strengthen the paper, there are areas for improvement. Firstly, the authors could provide more details on the methodology, including the sentiment analysis, multi-head attention mechanism, and model training. Secondly, ensuring the statistical significance of the findings would strengthen the validity of the results. Finally, discussing the potential applications and limitations of the model in a broader context would enhance the paper's relevance and impact.

---

## Round 0.2 · accepted · Accept

Congratulations. I have seen that you have addressed all comments of reviewers and my own feedback.

Reviewer 1 ·

Basic reporting

No comments

Experimental design

The description of the proposed method is still lacking. Might be suggested with minor revision.

Validity of the findings

No comments

Additional comments

No comments

Reviewer 3 ·

Basic reporting

no comment

Experimental design

no comment

Validity of the findings

no comment

Additional comments

no comment